# Occupancy of the American Three-Toed Woodpecker in a Heavily-Managed Boreal Forest of Eastern Canada

Vincent Lamarre [1] and Junior A. Tremblay [1,2,*]

1 Science and Technology Branch, Environment and Climate Change Canada, 1550 Avenue d'Estimauville, Québec, QC G1J 0C3, Canada; vincent.lamarre@canada.ca
2 Département des Sciences du Bois et de la Forêt, Université Laval, 2405 rue de la Terrasse, Québec, QC G1V 0A6, Canada
* Correspondence: junior.tremblay@canada.ca

**Abstract:** The southern extent of the boreal forest in North America has experienced intensive human disturbance in recent decades. Among these, forest harvesting leads to the substantial loss of late-successional stands that include key habitat attributes for several avian species. The American Three-toed Woodpecker, *Picoides dorsalis*, is associated with continuous old spruce forests in the eastern part of its range. In this study, we assessed the influence of habitat characteristics at different scales on the occupancy of American Three-toed Woodpecker in a heavily-managed boreal landscape of northeastern Canada, and we inferred species occupancy at the regional scale. We conducted 185 playback stations over two breeding seasons and modelled the occupancy of the species while taking into account the probability of detection. American Three-toed Woodpecker occupancy was lower in stands with large areas recently clear-cut, and higher in landscapes with large extents of old-growth forest dominated by black spruce. At the regional scale, areas with high probability of occupancy were scarce and mostly within protected areas. Habitat requirements of the American Three-toed Woodpecker during the breeding season, coupled with overall low occupancy rate in our study area, challenge its long-term sustainability in such heavily managed landscapes. Additionally, the scarcity of areas of high probability of occupancy in the region suggests that the ecological role of old forest outside protected areas could be compromised.

**Keywords:** boreal forest; clear-cutting; conservation; forest management; old-growth forest; *Picoides dorsalis*; protected areas

## 1. Introduction

The boreal forest represents about 48% of the world's forested biomes [1]. Boreal landscapes are shaped by natural disturbances such as windthrows, forest fires and insect outbreaks that generate a complex mosaic in vegetation structure and composition [2,3]. For several decades, however, the exploitation of natural resources has been adding to natural disturbances in this biome [4] and has modified the structure and composition of boreal ecosystems, including the reduction of late-successional stands called old-growth forests [5,6].

Old-growth boreal forests are characterized, among several attributes, by irregular vertical and horizontal structures, large volumes of deadwood either standing (snags) or fallen (coarse woody debris) and in different decaying stages [7]. These attributes are considered key for hundreds of species that depend upon dead or decaying woody material during some part of their life cycle and that are found disproportionately in old-growth forests [8,9]. The temporal continuity of these ecological attributes is also an essential characteristic for many species [10,11]. For instance, a recent study highlights the continuous supply of large slightly decayed snags in specific old-growth forest type as a key element to provide temporal stability in the foraging habitat of the Black-backed Woodpecker, *Picoides arcticus* [12].

Deadwood provides foraging or breeding substrate for several vertebrate species [13]. Boreal woodpeckers, for example, are "ecosystem engineers" that create nesting cavities for other vertebrates and are considered indicator species for deadwood-associated biodiversity [14,15]. Among woodpecker species found within the boreal biome in North America, the American Three-toed Woodpecker, *Picoides dorsalis*, is the most strongly associated with continuous old spruce forests at the landscape scale [16–18]. Harvesting, especially of old-growth coniferous forests, is thought to be detrimental to the species by limiting the supply of deadwood [16,18–20]. However, most of the studies on the species in eastern Canada occurred in regions where forests were harvested for the first time and where mature and old stands were still relatively abundant amongst residual forests. In heavily-managed forest landscapes, the persistence of the species remains uncertain. Indeed, in a recent attempt to gain insight on the breeding ecology of American Three-toed Woodpecker in heavily managed forest at the southern edge of its breeding range (New-Brunswick, Canada), Craig et al. [21] reported the species in only 5.9% of the playback stations, and found no predictors of site occupancy although most nests were found in recently dead black spruce trees.

Here, we assess the influence of habitat characteristics at the stand and landscape scales on the occupancy of American Three-toed Woodpecker during the breeding period in a heavily-managed, unburned boreal forest landscape of eastern Canada, and infer the probability of occupancy of the species at the regional scale. We expect that American Three-toed Woodpecker occupancy would be favoured by old spruce forest stands but negatively affected by recently harvested forest stands and higher forest fragmentation, and that the probability of occupancy of the species at the regional scale would be low.

## 2. Materials and Methods

### 2.1. Study Area

This study was conducted at the Forêt Montmorency (Université Laval's Experimental Forest), and the Réserve Faunique des Laurentides, Québec, Canada (47°4′ N. 71°0′ W.; Figure 1). The study area covers approximately 210 km$^2$ within the balsam fir-white birch bioclimatic domain in the continuous boreal forest subzone [22], which represents the southernmost section of the boreal forest in eastern Canada [23]. More precisely, the study area is classified within the high-elevation balsam fir–white birch zone with elevation ranging between 600 and 1100 m [24]. Forest stands within the study area are dominated by balsam fir *Abies balsamea* and black spruce *Picea mariana*, the latter being more abundant at higher elevation. Companion species are white birch (*Betula papyrifera*), white spruce (*Picea glauca)*, and tamarack *(Larix laricina)*. In the region, average age-class structure of the natural variability observed over the last few centuries includes 86% of old forest stands [25]. However, intensive logging during the 20th and early 21st century led to a net reduction in the cover of mature and old forest stands [26]. During the 1909–2005 time frame, 105% of the high-elevation boreal landscape has been harvested with some areas logged twice [27]. Consequently, the studied area is now dominated by young regenerating stands (0–19 years; 39%), while the cover of old forests (>90 years) is approximately 25%. Young closed (20–59 years) and mature (60–89 years) forests as well as non-forested lands (e.g., lakes, wetlands, etc.) respectively cover 11, 16 and 9% of the study area. Dominant cover tree species of old forest stands are balsam fir (55%), black spruce (42%) and tamarack (3%).

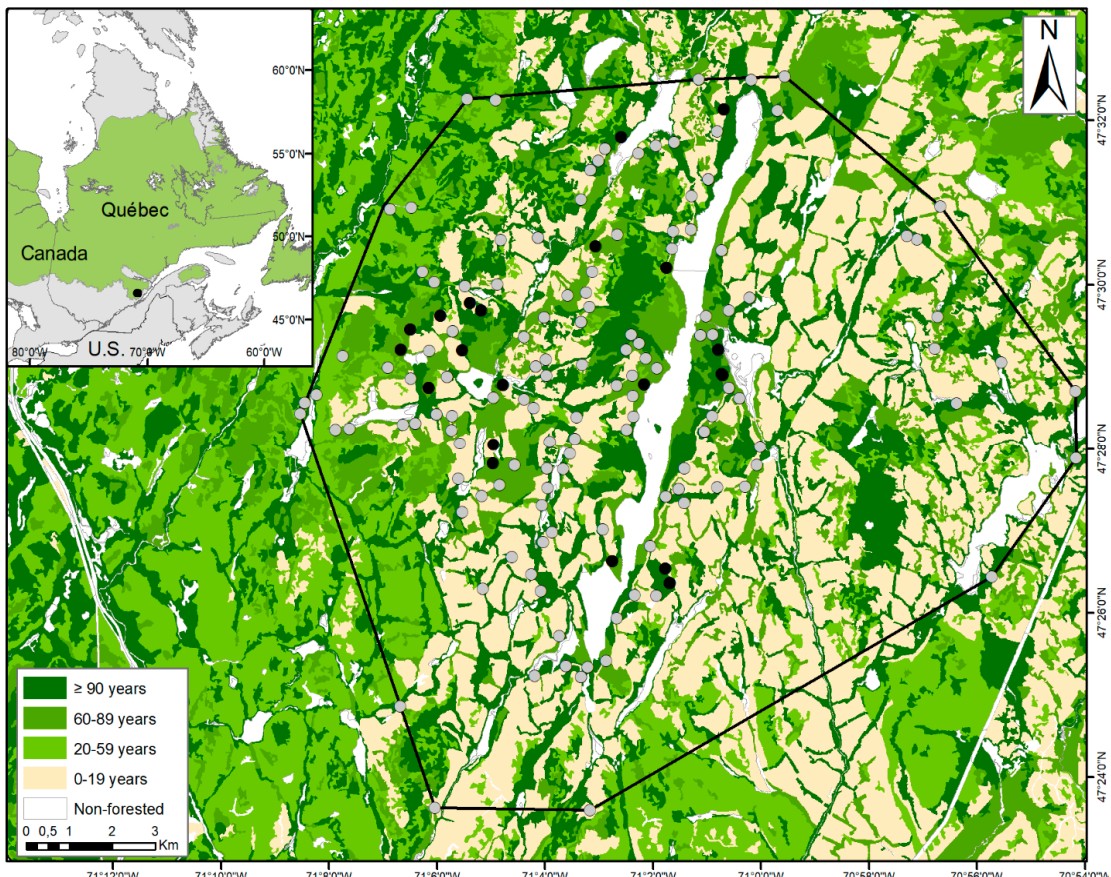

**Figure 1.** Habitat composition and location of surveyed stations where American Three-toedWoodpecker was detected (black circles) and not detected (grey circles) during occupancy surveys conducted between 2016 and 2017 within the Forêt Montmorency and the Réserve Faunique des Laurentides (Québec, Canada). Boundaries of the boreal zone [28] are delimited by the green zone in the left insert. The black line delineates the boundaries of the study area in the large insert.

### 2.2. Woodpecker Surveys

In 2016 and 2017, we conducted woodpecker occupancy surveys during American Three-toed Woodpecker breeding season at stationary stations distributed along forest roads. We did not stratify the sampling *per se* but rather targeted areas with clusters of unharvested patches within the vicinity of old forest stands. Surveyed stations were distanced by a minimum of 255 m (mean ± standard deviation: 554 ± 442 m; range 255–3414 m), while mean elevation at stations was 889 ± 37 m (range 819–1015 m). Surveys lasted five minutes (300 s), during which conspecific playbacks were displayed, and we noted the time elapsed since the start of the playback when an individual was observed or heard. Similar to the method used by Craig et al. [21], the 300-s conspecific display was divided into two 120-s observation periods which constituted the two visits used in occupancy analyses (see below). Both observation periods were interspersed by a 60-s pause. Playbacks were mixed from recordings obtained from xeno-canto [29] and the Macaulay Library [30]. Playbacks could be heard to approximately 450 m by the human ear in the field. We conducted surveys between 5:47 a.m. and 3:32 p.m., when precipitations were absent or minimal and wind speed ≤ 3 on the Beaufort scale (12–19 km h$^{-1}$). Afterward, we accessed three-hour mean wind speed from the nearest weather station located 18 km northwest of the study area for further analysis [31].

### 2.3. Habitat Characteristics

We investigated the influence of habitat characteristics that were likely to affect the occupancy of American Three-toed Woodpecker by calculating vegetation covariates within buffers of 250 and 750 m-radii centered on each surveyed station, representing respectively the stand and landscape scales. The landscape scale was selected based on home range size estimates obtained for one female and two male American Three-toed Woodpeckers during the nesting period in the study area, which averaged 177 ha (J.A. Tremblay, unpublished data). Vegetation covariates were calculated using the eco-forestry layers available as of 2017 for the study area [32]. At the stand scale, we calculated the area (ha) covered by recent clear-cut stands *clear_cut*. At the stand and landscape scales, we measured the area (ha) covered by old forest (≥90 years) dominated by black spruce *old_spruce* and mean forest age *mean_age*. We weighted *mean_age* by the area covered by even-aged stands within the buffers around each surveyed station. We calculated the standard deviation of mean stand age *sd_age* as an index of habitat fragmentation at the landscape scale. Based on the literature in eastern Canada [16–18], the area covered by non-forested lands in the landscape, the area covered by young (20–59 years) and mature (60–89 years) forests at the stand and landscape scales have been considered marginal for the American Three-toed Woodpecker in the selection of its habitat during the breeding season and thus were excluded from occupancy analyses (see below). We nevertheless compared mean value of all habitat characteristics between stations where the American Three-toed Woodpecker was detected, not detected and a set of 185 random stations distributed randomly into the study (Table 1) area using a Kruskal–Wallis test in R statistical environment version 4.0.2 [33].

**Table 1.** Description and mean values ± standard deviation (s.d.) of habitat characteristics measured at the stand (250 m-radii) and landscape (750 m-radii) scales at stations where the American Three-toed Woodpecker was detected, not detected and at random stations during occupancy surveys in managed boreal forest in eastern Canada. Shared letters indicate no significant differences in habitat characteristics among stations (Kruskal–Wallis test). Habitat characteristics in bold are included in occupancy analyses.

| Habitat Characteristic | Description | Not Detected (n = 163) | Detected (n = 22) | Random (n = 185) |
|---|---|---|---|---|
| **clear_cut_250** | Area (ha) covered by recent (0–19 years) clear-cut | 6.58 (4.62) [a] | 3.91 (4.17) [b] | 6.60 (5.89) [a] |
| **mean_age_250** | Mean forest age (year) | 70.78 (25.44) [a] | 83.27 (30.27) [a] | 64.92 (32.34) [b] |
| **mean_age_750** | | 70.07 (15.34) [a] | 76.27 (20.63) [a] | 64.33 (19.47) [b] |
| young_250 | Area (ha) covered by young closed forest (20–59 years) | 1.29 (2.96) [a] | 0.86 (1.54) [a,b] | 3.24 (5.06) [b] |
| young_750 | | 14.71 (24.20) [a] | 8.34 (10.62) [a] | 27.03 (35.24) [b] |
| mature_250 | Area (ha) covered by mature forest (60–89 years) | 4.87 (4.64) [a] | 6.23 (5.72) [a] | 3.11 (3.80) [b] |
| mature_750 | | 38.49 (21.80) [a] | 48.89 (23.05) [a] | 28.00 (22.32) [b] |
| old_spruce_250 | Area (ha) covered by old forest (≥90 years) dominated by black | 2.92 (3.04) [a] | 3.85 (3.46) [a] | 2.26 (3.51) [b] |
| **old_spruce_750** | spruce | 21.87 (13.18) [a] | 26.00 (16.33) [a] | 19.92 (15.08) [a] |
| **sd_age_750** | Standard deviation of forest age | 56.71 (8.89) [a] | 53.08 (9.35) [a] | 54.62 (9.80) [a] |
| nf_750 | Area (ha) covered by non-forested lands in the landscape | 21.04 (19.98) [a] | 26.44 (23.93) [a] | 12.00 (16.16) [b] |

### 2.4. Occupancy Analyses

We used single-species occupancy modeling in the *unmarked* R library to investigate the influence of habitat characteristics on the probability of occupancy of American Three-toed Woodpecker [34,35]. We converted survey detections of American Three-toed Woodpecker into presence–absence data. To account for imperfect detection probability during surveys, we first estimated the effect of year, Julian day, wind speed (m s$^{-1}$) and hour of the day on detection probability of American Three-toed Woodpecker. We used a two-step approach [36] to determine which detection parameter(s) to retain in the occupancy models. We first estimated the effect of each detection parameter by ranking univariate models in which occupancy was held constant. We also included a null model in which detection and occupancy were held constant (Table A1). The five candidate models were ranked based

on the second-order Akaike's information criterion AICc [37]; using the aictab function of the *AICcmodavg* R library [38].

We developed a set of 10 biologically relevant candidate occupancy models representing hypotheses to investigate the influence of habitat characteristics on the probability of occupancy of American Three-toed Woodpecker (Table 2). This set also included a null model of constant occupancy. Strongly correlated covariates ($|r| \geq 0.60$; Table A2) were not included in the same candidate model to reduce multicollinearity. We ranked candidate models based on the $AIC_c$ and we made inference on the top models with $\Delta AICc < 2$. Parameter-averaged predictions of covariates appearing in the most parsimonious models were calculated over their measured range while holding other variables at their mean value. Finally, we inferred a predicted probability of occupancy of American Three-toed Woodpecker from the top models (Table 2) at the regional scale (30 km $\times$ 30 km square zone) within the balsam fir–white birch bioclimatic domain.

**Table 2.** Candidates models, number of parameters (*k*), second-order Akaike's information criterion (AICc), ΔAICc, Akaike weights (*ω*), log-likelihood (LL) of the candidate models assessing occupancy (ψ) and detection probability (*p*) of American Three-toed Woodpecker in a managed boreal forest in eastern Canada.

| Candidate Model | k | $AIC_c$ | $\Delta AIC_c$ | ω | LL |
|---|---|---|---|---|---|
| ~ *p*(time) ~ ψ(clear_cut_250 + old_spruce_750) | 5 | 176.56 | 0.00 | 0.49 | −83.11 |
| ~ *p*(time) ~ ψ(clear_cut_250) | 4 | 178.46 | 1.90 | 0.19 | −85.12 |
| ~ *p*(time) ~ ψ(mean_age_250 + sd_age_750) | 5 | 178.99 | 2.43 | 0.15 | −84.33 |
| ~ *p*(time) ~ ψ(clear_cut_250 + mean_age_750) | 5 | 180.22 | 3.65 | 0.08 | −84.94 |
| ~ *p*(time) ~ ψ(mean_age_250) | 4 | 182.46 | 5.89 | 0.03 | −87.12 |
| ~ *p*(time) ~ ψ(old_spruce_250 + sd_age_750) | 5 | 182.76 | 6.20 | 0.02 | −86.21 |
| ~ *p*(time) ~ ψ(sd_age_750) | 4 | 183.63 | 7.06 | 0.01 | −87.70 |
| ~ *p*(time) ~ ψ(mean_age_250 + old_spruce_750) | 5 | 183.86 | 7.30 | 0.01 | −86.76 |
| ~ *p*(time) ~ ψ(null) | 3 | 185.05 | 8.49 | 0.01 | −89.46 |
| ~ *p*(time) ~ ψ(old_spruce_250) | 4 | 185.25 | 8.68 | 0.01 | −88.51 |

## 3. Results

Between 19 May and 13 July 2016 and between 26 May and 29 June 2017, we broadcasted American Three-toed Woodpecker playbacks at 185 stations, including 35 revisited stations in 2017. American Three-toed Woodpecker was detected at 22 (11.9%) of the 185 stations. Forest composition and structure differed between sampled and random stations since we targeted our sampling effort towards clusters of older residual forest patches distributed close to non-forested lands such as waterbodies and wetlands in the study area (Table 1). Indeed, mean forest age at the stand and landscape scale and the area covered by old spruce at the stand scale were lower at random than sampled stations. In addition, the area covered by mature stands at both scales was also lower at random stations, while the dominance of young stands tended to be greater. Finally, non-forested habitats in the landscape were less important at random than sampled stations (Table 1).

Time of the day was retained in the most parsimonious model and accounted for 42% of model selection weight (Table A1). The models including year ($\Delta AIC_c$ = 1.61) and date ($\Delta AIC_c$ = 1.95) were equivalent and did not differ from the null model ($\Delta AIC_c$ = 1.77). Only time of the day influenced the probability of detection of American Three-toed Woodpecker ($p_{time}$: 0.37, 95% C.I.: [0.03, 0.71]) and therefore was the only detection parameter included in occupancy models. Mean occupancy rate of American Three-toed Woodpecker in the study site was 0.17 ± 0.04, and mean detection probability when the time of the day was fixed to its mean value was 0.48 ± 0.11.

Two models had a substantial level of empirical support in influencing occupancy of the American Three-toed Woodpecker ($\Delta AIC_c < 2$; Table 2). Together, these models accounted for 68% of the AIC weight. At the stand scale, occupancy of American Three-toed Woodpecker decreased with increasing recent clear-cut area ($\psi_{clear\_cut\_250}$: $-0.20$, 95% C.I.: [$-0.34$, $-0.06$], Figure 2a). The area covered by recent clear-cuts at the stand scale was also lower at stations where American Three-toed Woodpecker was detected compared to stations where the species was not detected or random stations (Table 1). At the landscape scale, occupancy of the species tended to increase with an increase in the area covered by old forest dominated by black spruce ($\psi_{old\_spruce\_750}$: 0.04, 95% C.I.: [0.00–0.08], Figure 2b), with increasing uncertainty in confidence intervals > 40 ha most likely due to the scarcity of large old spruce forest stands in our study area.

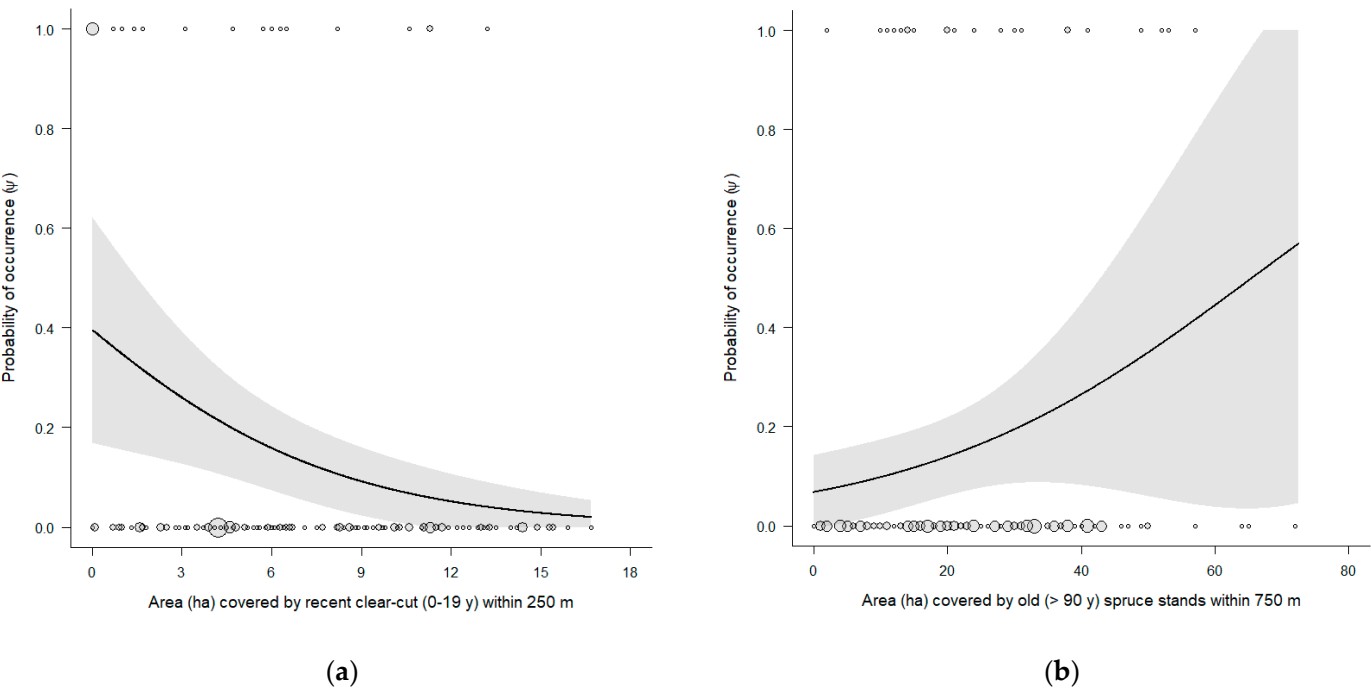

(**a**)            (**b**)

**Figure 2.** Influence on the occupancy of American Three-toed Woodpecker in managed boreal forest in eastern Canada of the area (ha) covered by (**a**) recent clear-cut stands at the stand scale (250 m radii) and (**b**) old spruce stands at the landscape scale (750 m radii). The shaded grey area represents 95% confidence interval, and symbols represent the distribution of raw data with symbol size proportional to the (log + 1) number of observations.

Extrapolating results inferred from our top occupancy models (Table 2) to a larger extent within the balsam fir–white birch bioclimatic domain showed that only 12.3% of the region had moderate- to high-predicted probability of occupancy (>0.5) of the American Three-toed Woodpecker. About half of these areas (6.7%) were located within protected areas. Areas with high-predicted probability of occupancy (>0.75) of the species represented 4.7% of the region where only 1.0% were in managed forests outside protected areas (Figure 3).

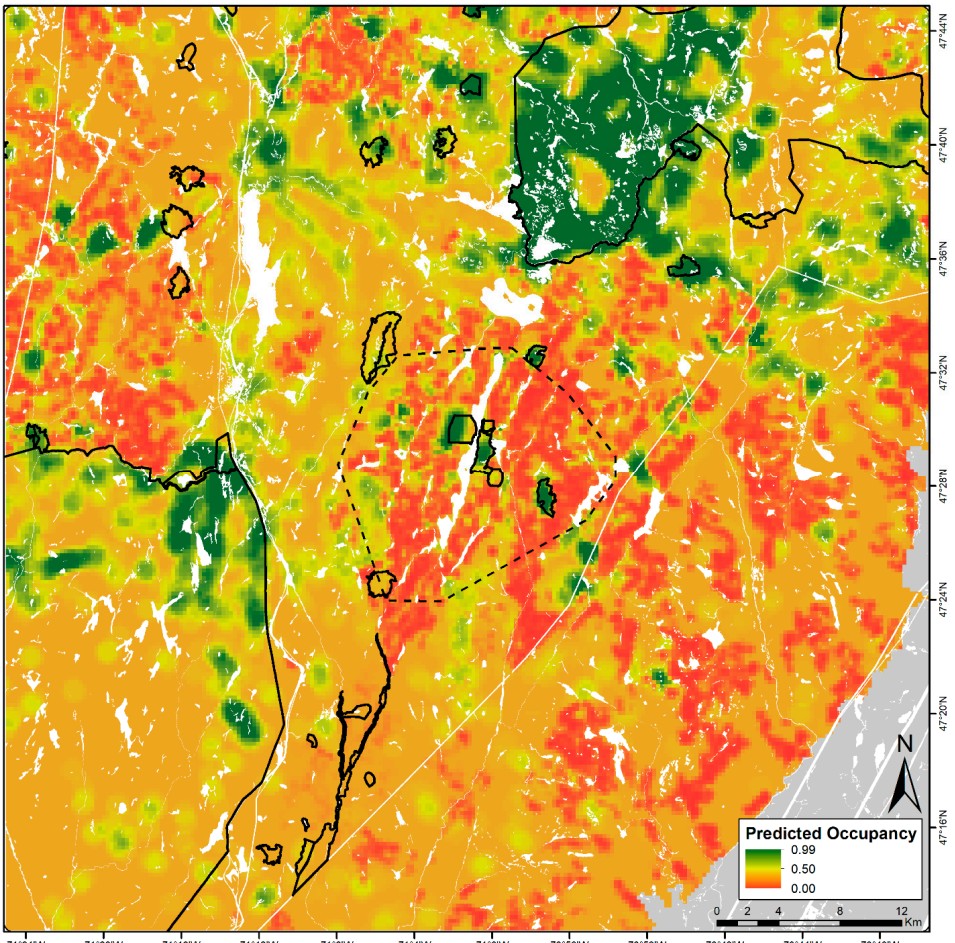

**Figure 3.** Predicted probability of occupancy of American Three-toed Woodpecker at the regional scale within the balsam fir–white birch bioclimatic domain. Probability of occupancy is inferred from the top-ranking models ($\Delta AIC_c < 2$; Table 2). The dashed and solid black lines respectively delineate the boundaries of the study area and protected areas and white polygons indicate non-forested lands.

## 4. Discussion

In a heavily-managed landscape at the southern edge of the boreal forest, the American Three-toed Woodpecker exhibited a low mean occupancy rate ($0.17 \pm 0.04$) where its probability of occupancy decreased rapidly with increasing area of recent clear-cut at the stand scale. The amount of old spruce forest was positively associated with species occupancy at the landscape scale. Our results suggest that logging history in the region may have created an unsuitable forest landscape (i.e., dominated by younger age-classes and fragmented residual older forest stands) to support a long-term population of American Three-toed Woodpecker.

Habitat associations of the American Three-toed Woodpecker vary across its range. In a meta-analysis focusing on successional trajectories of bird communities following fire and harvesting in boreal forests of western Canada, Schiek and Song [39] report the American Three-toed Woodpecker more common in older mixed wood and white spruce. In western Quebec, Imbeau and Desrochers [16] document a positive association between the American Three-toed Woodpecker with the amount of old spruce forest in continuous coniferous forest. Similarly, Cadieux and Drapeau [18] report a higher occurrence of the species in black spruce forest stands older than 90 years and no occurrence in younger coniferous stands. Accordingly, we find that the occupancy of the American Three-toed Woodpecker during the breeding period increases with old spruce forest stands in a managed landscape. This result agrees with our prediction as it has been shown that the

American Three-toed Woodpecker preferentially forages on relatively large and senescent or recently dead coniferous trees to access bark-associated beetles (Scolytinae) by scaling layers of bark with strong preference for black spruce in eastern Canada [17,40,41].

Recent clear-cut at the stand scale have a negative effect on the occupancy of the American Three-toed Woodpecker. This result agrees with our expectation and with previous studies about the sensitivity of the species to forest harvesting, especially in the eastern part of its distribution range [19,20,41,42]. Accordingly, in the black spruce moss domain of eastern Canada, recent clear-cuts have a lower density of snags, and the American Three-toed Woodpecker is solely found in old-growth forest [19]. In addition to a reduction in the deadwood abundance, long-term recruitment of large-diameter snags is compromised in clear-cut stands [43]. In a closely-related species, the Black-backed Woodpecker, old coniferous forests with a greater volume of recently dead trees than adjacent recent cuts are selected for foraging during the breeding period [44]. Similar foraging avoidance of recent clear-cut areas may be occurring in our study site for the American Three-toed Woodpecker. Indeed, timber harvesting peaked during the period 1996–2004 and ended in 2009, and although we did not quantify the volume and decay stages of deadwood, it is reasonable to think that most deadwood in clear-cuts had entered late decay classes and was of low quality for foraging American Three-toedWoodpecker at the time of our study.

Our results do not report a predictive effect of habitat fragmentation at the landscape scale on the occupancy of the American Three-toed Woodpecker. Forest edge avoidance by foraging American Three-toed Woodpecker has been reported, where high-quality substrates near stand edges are used less frequently than available [41]. In addition, movements of foraging woodpeckers also appear to be constrained in residual forests following harvesting [16,41]. Hence, the effect of habitat fragmentation seems to act on a finer scale, and our results suggest that at a larger scale (i.e., landscape), the amount of old forests may strongly influence the occurrence of the species. For instance, most of our detection of the species occurred close to forest patches of mature or old forests rather than residual strips of forest (Figure 1).

Only 1% of the forest stands in managed forests at the regional scale show high predicted probability of occupancy of the American Three-toed Woodpecker. This is likely a consequence of forestry practices, mainly driven by clear-cutting during the last century which has led to a substantial reduction of old forest cover in the region [26,27]. Such practices can hardly sustain biodiversity associated with old coniferous forest stands, especially for species with large home ranges. For example, habitat alteration from anthropogenic activities is a threat of high concern for populations of Woodland Caribou *Rangifer tarandus caribou* across Canada's boreal biomes, including the local population in our study area, which is considered "not self-sustaining" [45]. Fennoscandia previously experienced a similar situation where old-growth forests have almost completely disappeared [46,47], and it is estimated that 30–50% of the red-listed species in these regions are associated with old-growth forest attributes [48,49]. In boreal ecosystems, conservation strategies cannot be based solely on a network of protected areas but rather on how unprotected areas are managed [50]. Hence, management practices mimicking the attributes of old-growth forests by ensuring a continuous recruitment of deadwood appear important for maintaining species associated with old spruce forests. Within these practices, partial harvesting may be efficient in maintaining a relatively high abundance of deadwood and associated deadwood-dependent species such as the American Three-toed Woodpecker [51,52], although the efficiency of such practices has not yet been assessed. The ecological role of old forests in our study area seems to be altered, and the situation may require revised management practices and a passive restoration of the ecological integrity of old forests outside protected areas in the region *sensu* [53].

## 5. Conclusions

With 185 stations sampled over two breeding seasons, our study on the occupancy of the American Three-toed Woodpecker is one of the first conducted in a heavily harvested landscape at the southern edge of the boreal forest. The overall low occupancy of the species within our study area raises questions about the long-term sustainability of such heavily managed landscapes for the American Three-toed Woodpecker. Areas of high-predicted occupancy of the species in the studied region are mostly found in protected areas, providing evidence that heavy forest harvesting is a detrimental driver that likely contributed to the significant long-term declining trends of the species over the 1970–2019 period in the province of Québec ($-0.96\%.\text{year}^{-1}$; 95% C.I.: $[-0.70--1.22]$; [54]) and at a larger scale in the boreal hardwood transition bird conservation region ($-3.5\%.\text{year}^{-1}$; 95% C.I.: $[-2.0--5.2]$; [55]). The southern part of the boreal forest in eastern Canada, where we conducted our study, has experienced one of the most important human disturbances in the past decades, mainly related to forest harvesting [4]. We pledge for more detailed studies on this discrete keystone species in different regions with varying forest harvesting intensity, at the stand scale (going from clear-cutting to partial harvesting) and landscape scale (from pristine to heavily harvested), focusing on demographic parameters (i.e., reproductive success and survival) and habitat selection.

**Author Contributions:** Conceptualization, J.A.T.; methodology, J.A.T.; formal analysis, J.A.T., V.L.; investigation, J.A.T., V.L.; resources, J.A.T.; data curation, V.L.; writing—original draft preparation, V.L.; writing—review and editing, J.A.T., V.L.; visualization, V.L.; supervision, J.A.T.; project administration, J.A.T.; funding acquisition, J.A.T. All authors have read and agreed to the published version of the manuscript.

**Funding:** This research was funded by Environment and Climate Change Canada.

**Institutional Review Board Statement:** Not applicable.

**Informed Consent Statement:** Not applicable.

**Data Availability Statement:** The data presented in this study are openly available in https://doi.org/10.5061/dryad.9kd51c5gc.

**Acknowledgments:** We are grateful to André Desrochers for sharing the code to calculate habitat characteristics and to Mark J. Mazerolle for advice related to statistical analyses. We are also thankful to the personnel at the Forêt Montmorency for their support and to Michel Robert for the loan of all-terrain vehicles. Finally, we thank Francis Lessard and Julien St-Amand for their help with fieldwork.

**Conflicts of Interest:** The authors declare no conflict of interest. The funders had no role in the design of the study; in the collection, analyses, or interpretation of data; in the writing of the manuscript, or in the decision to publish the results.

## Appendix A

**Table A1.** Model selection based on the $\text{AIC}_c$ for the estimation of detection probability ($p$) of American Three-toed Woodpecker in managed boreal forest in eastern Canada. The number of parameters ($k$), second-order Akaike's information criterion ($\text{AIC}_c$), $\Delta\text{AIC}_c$, Akaike weights ($\omega$), log-likelihood (LL) are included in the table and occupancy ($\psi$) is held constant for each candidate model.

| Candidate Model | $k$ | $\text{AIC}_c$ | $\Delta\text{AIC}_c$ | $\omega$ | LL |
|---|---|---|---|---|---|
| ~ $p$(time) | 3 | 185.05 | 0.00 | 0.42 | −89.46 |
| ~ $p$(year) | 3 | 186.66 | 1.61 | 0.19 | −90.27 |
| ~ $p$(null) | 2 | 186.82 | 1.77 | 0.17 | −91.38 |
| ~ $p$(date) | 3 | 187.00 | 1.95 | 0.16 | −90.44 |
| ~ $p$(wind) | 3 | 188.60 | 3.55 | 0.07 | −91.23 |

**Table A2.** Correlation among habitat characteristics included in occupancy analyses.

|  | mean_age_250 | clear_cut_250 | old_spruce_250 | mean_age_750 | sd_age_750 | old_spruce_750 |
|---|---|---|---|---|---|---|
| mean_age_250 | 1 |  |  |  |  |  |
| clear_cut_250 | −0.53 | 1 |  |  |  |  |
| old_spruce_250 | 0.42 | 0.03 | 1 |  |  |  |
| mean_age_750 | 0.66 | −0.36 | 0.19 | 1 |  |  |
| sd_age_750 | 0.19 | 0.31 | 0.32 | 0.28 | 1 |  |
| old_spruce_750 | 0.29 | 0.17 | 0.60 | 0.37 | 0.23 | 1 |

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
