# Peer review of "Occupancy of the American Three-Toed Woodpecker in a Heavily-Managed Boreal Forest of Eastern Canada"

_diversity, doi:10.3390/d13010035_

Round 1

Reviewer 1 Report

Dear Editor,

I read the manuscript ‘Occupancy of the American Three-toed Woodpecker in a Heavily Managed Boreal Forest of Eastern Canada’ with interest. The authors conducted a habitat selection study, using occupancy analysis, to assess stand and landscape characteristics of an intensively managed forest landscape. I consider it a good quality study, well presented and for which I raised only some minor concerns. Anyway, I would like the authors to seriously consider my concerns about the methods. Perhaps, the assumptions for occupancy analysis are not respected and other techniques may be better suited for data analysis (e.g. GLMM). In addition, I suggest the authors to improve the introduction and the discussion (for which I provide some suggestions below), in order to include the study in a broader perspective. I consider that the authors should go beyond the single-species focus (as they do in one case in the discussion), if they wish to publish in a journal called ‘Diversity’.

L82: Please mention the year of the layer explicitly.

L99: Was the sampling design stratified among forest age classes? If yes, how many points per class? Or if it was random, what was the final distribution od point per classes? Please, add more details to this or previous section about the sampling design.

L106: It is not clear how long did each playback last. Was the playback calling for 2 minutes, then a 1-minute pause, and on again for other 2 minutes? Or did I misinterpret? How did the authors record an individual? The recorded calling individuals only during the 2 -minute playback session? How could they distinguish between 2 individuals calling in such short time? Please, provide more details on the sampling technique.

L127-130: Perhaps this sentence needs to be rephrased.

L141: Did the authors checked whether their sampling design respected all the necessary assumptions for occupancy modelling? Can they be sure about the independence between surveys, considering the short time period separating the two-surveys? Also, detectability modelling is best suited for sampling design including 3 or more replicate surveys. Did the authors use other ways to minimise the detectability error, considering that they planned only two replicate surveys?

L141: Please, include a citation for the library ‘unmarked’.

L260-275: This part could be expanded to include a more broad and transcontinental perspective of forests, woodpeckers and management, I provide some references but more can be found:

  1. Basile M, Asbeck T, Pacioni C, Mikusiński G, Storch I. 2020 Woodpecker cavity establishment in managed forests: relative rather than absolute tree size matters. Wildlife Biol. 2020. (doi:10.2981/wlb.00564)
  2. Cockle KL, Martin K, Wesołowski T. 2011 Woodpeckers, decay, and the future of cavity‐nesting vertebrate communities worldwide. Front. Ecol. Environ. 9, 377–382. (doi:10.1890/110013)
  3. Mikusiński G, Gromadzki M, Chylarecki P. 2001 Woodpecker as indicator of forest bird diversity. Conserv. Biol. 15, 208–217.
  4. Angelstam P, Mikusiński G. 1994 Woodpecker assemblages in natural and managed boreal and hemiboreal forest - a review. Ann. Zool. Fennici. 31, 157–172.
  5. Puverel C, Abourachid A, Böhmer C, Leban J, Svoboda M, Paillet Y. 2019 This is my spot: What are the characteristics of the trees excavated by the Black Woodpecker? A case study in two managed French forests. For. Ecol. Manage. 453, 117621. (doi:10.1016/j.foreco.2019.117621)
  6. Walczak Ł, Kosiński Z, Stachura-Skterczyńska K. 2013 Factors affecting the occurrence of Middle Spotted Woodpeckers as revealed by forest inventory data. Balt. For. 19, 81–88.
  7. Bütler R, Angelstam P, Ekelund P, Schlaepfer R. 2004 Dead wood threshold values for the three-toed woodpecker presence in boreal and sub-Alpine forest. Biol. Conserv. 119, 305–318. (doi:10.1016/j.biocon.2003.11.014)

Author Response

I read the manuscript ‘Occupancy of the American Three-toed Woodpecker in a Heavily Managed Boreal Forest of Eastern Canada’ with interest. The authors conducted a habitat selection study, using occupancy analysis, to assess stand and landscape characteristics of an intensively managed forest landscape. I consider it a good quality study, well presented and for which I raised only some minor concerns. Anyway, I would like the authors to seriously consider my concerns about the methods. Perhaps, the assumptions for occupancy analysis are not respected and other techniques may be better suited for data analysis (e.g. GLMM). In addition, I suggest the authors to improve the introduction and the discussion (for which I provide some suggestions below), in order to include the study in a broader perspective. I consider that the authors should go beyond the single-species focus (as they do in one case in the discussion), if they wish to publish in a journal called ‘Diversity’.

***We thank the reviewer for his helpful comments. We address each specific comments below.

L82: Please mention the year of the layer explicitly.

***We added the year of the layer (2017) as suggested at l.123-124. We also removed the reference to eco-forestry layers at l.81 as we judged unnecessary to refer to it anymore.

L99: Was the sampling design stratified among forest age classes? If yes, how many points per class? Or if it was random, what was the final distribution od point per classes? Please, add more details to this or previous section about the sampling design.

***We did not stratify our sampling per se but rather sampled as much as we can areas with old forest stands nearby. We added these details at l.101-102.

L106: It is not clear how long did each playback last. Was the playback calling for 2 minutes, then a 1-minute pause, and on again for other 2 minutes? Or did I misinterpret? How did the authors record an individual? The recorded calling individuals only during the 2 -minute playback session? How could they distinguish between 2 individuals calling in such short time? Please, provide more details on the sampling technique.

***We acknowledge that these sentences lack of clarity and we rephrased the methods related to playback and observation periods at l.104-109. We also added a reference at l.107 (Craig, C.; Mazerolle, M.J.; Taylor, P.D.; Tremblay, J.A.; Villard, M.-A. Predictors of Habitat Use and Nesting Success for Two Sympatric Species of Boreal Woodpeckers in an Unburned, Managed Forest Landscape. Forest. Ecol. Manag. 2019, 438, 134–141, doi:10.1016/j.foreco.2019.02.016) as we used a similar approach to conduct playback surveys. Finally, we specified at l.146-147 that analyses are based on presence-absence data from our playback surveys.

L127-130: Perhaps this sentence needs to be rephrased.

***We rephrased the sentence as suggested (l.127-128).

L141: Did the authors checked whether their sampling design respected all the necessary assumptions for occupancy modelling? Can they be sure about the independence between surveys, considering the short time period separating the two-surveys? Also, detectability modelling is best suited for sampling design including 3 or more replicate surveys. Did the authors use other ways to minimise the detectability error, considering that they planned only two replicate surveys?

***We are confident our approach is the right one, and we added a reference at l.107 (Craig, C.; Mazerolle, M.J.; Taylor, P.D.; Tremblay, J.A.; Villard, M.-A. Predictors of Habitat Use and Nesting Success for Two Sympatric Species of Boreal Woodpeckers in an Unburned, Managed Forest Landscape. Forest. Ecol. Manag. 2019, 438, 134–141, doi:10.1016/j.foreco.2019.02.016) to a recently published paper that used a method similar to ours to support the choice of occupancy modelling.

L141: Please, include a citation for the library ‘unmarked’.

***We added a reference to the unmarked package as suggested (Fiske, I.; Chandler, R. unmarked : An R Package for Fitting Hierarchical Models of Wildlife Occurrence and Abundance. J. Stat. Soft. 2011, 43, doi:10.18637/jss.v043.i10). Please see ref #35, l.146.

L260-275: This part could be expanded to include a more broad and transcontinental perspective of forests, woodpeckers and management, I provide some references but more can be found:

  1. Basile M, Asbeck T, Pacioni C, Mikusiński G, Storch I. 2020 Woodpecker cavity establishment in managed forests: relative rather than absolute tree size matters. Wildlife Biol. 2020. (doi:10.2981/wlb.00564)
  2. Cockle KL, Martin K, Wesołowski T. 2011 Woodpeckers, decay, and the future of cavity‐nesting vertebrate communities worldwide. Front. Ecol. Environ. 9, 377–382. (doi:10.1890/110013)
  3. Mikusiński G, Gromadzki M, Chylarecki P. 2001 Woodpecker as indicator of forest bird diversity. Conserv. Biol. 15, 208–217.
  4. Angelstam P, Mikusiński G. 1994 Woodpecker assemblages in natural and managed boreal and hemiboreal forest - a review. Ann. Zool. Fennici. 31, 157–172.
  5. Puverel C, Abourachid A, Böhmer C, Leban J, Svoboda M, Paillet Y. 2019 This is my spot: What are the characteristics of the trees excavated by the Black Woodpecker? A case study in two managed French forests. For. Ecol. Manage. 453, 117621. (doi:10.1016/j.foreco.2019.117621)
  6. Walczak Ł, Kosiński Z, Stachura-Skterczyńska K. 2013 Factors affecting the occurrence of Middle Spotted Woodpeckers as revealed by forest inventory data. Balt. For. 19, 81–88.
  7. Bütler R, Angelstam P, Ekelund P, Schlaepfer R. 2004 Dead wood threshold values for the three-toed woodpecker presence in boreal and sub-Alpine forest. Biol. Conserv. 119, 305–318. (doi:10.1016/j.biocon.2003.11.014)

***We improved this section of the discussion l.271-277 to add more specific information about other species (i.e. Woodland Caribou) and include transcontinental perspective of old-growth forests as suggested.

Reviewer 2 Report

This manuscript titled, “Occupancy of the American Three-toed Woodpecker in a heavily managed boreal forest of Eastern Canada”, used playback surveys to model the occupancy in a heavily managed landscape. It is an interesting, good written and important manuscript and it highlight that conservation strategies should focus more the management strategies of unprotected areas. I have only a few minor comments which are listed below:

Line 86. Add that some forests had been logged twice, and that this is equivalent of more than 105%. Additional remove the space in “105 %”.

Line 100-102. From the results I understand that you did not surveyed the same station over the years, no repeated measurements. It will be good to clarify this here.

Line 127-130. It need to better explained why these variable were excluded, percentage of for example mature forest may be an important predictor when old forest is not present.

Table 1 Why did you measure clear cut only at the stand level while in the text you mention that you measured at both levels?

Table 1. Habitat characteristic column in the table can be simplified by removing _250 and _750 from the text. The third column (“Scale”) give the same information!

Line 167-169. See previous comment! It need to be clarified in the method if you recorded different station during the survey years or that some were surveyed twice.

192-194. There is indeed a tendency of increase but confidence interval are large and thus a lot of uncertainty! This makes it questionable how well this variable will predict woodpecker occurrence, so this should be treated with care and may be discussed in the discussion.

Author Response

This manuscript titled, “Occupancy of the American Three-toed Woodpecker in a heavily managed boreal forest of Eastern Canada”, used playback surveys to model the occupancy in a heavily managed landscape. It is an interesting, good written and important manuscript and it highlight that conservation strategies should focus more the management strategies of unprotected areas. I have only a few minor comments which are listed below:

Line 86. Add that some forests had been logged twice, and that this is equivalent of more than 105%. Additional remove the space in “105 %”.

***Done as suggested (l.84-86).

Line 100-102. From the results I understand that you did not surveyed the same station over the years, no repeated measurements. It will be good to clarify this here.

***We indeed revisited some stations in 2017, and we added this info at l.172-173. We did not considered these as repeated measurements, but we tested for the year effect on the probability of detection of the species (see l.147-149).

Line 127-130. It need to better explained why these variable were excluded, percentage of for example mature forest may be an important predictor when old forest is not present.

***We made the selection of variables in the model selection based on the species literature; we added this information and reworded these lines to clarify it (l. 130-134).

Table 1 Why did you measure clear cut only at the stand level while in the text you mention that you measured at both levels?

***Excellent observation. We indeed measured clear cut only at the stand level and we made the changes accordingly in the text (l.124-125).

Table 1. Habitat characteristic column in the table can be simplified by removing _250 and _750 from the text. The third column (“Scale”) give the same information!

***Good point, we removed the “Scale” column but kept “_250” and “_750” in Table 1 (l.142) to remain consistent with the nomenclature used in Table 2.

Line 167-169. See previous comment! It need to be clarified in the method if you recorded different station during the survey years or that some were surveyed twice.

***Please see above our response to the previous comment about revisited stations.

192-194. There is indeed a tendency of increase but confidence interval are large and thus a lot of uncertainty! This makes it questionable how well this variable will predict woodpecker occurrence, so this should be treated with care and may be discussed in the discussion.

***Indeed, uncertainty is large for > 40 ha of old spruce forest as seen in Fig.2b due to the scarcity of these stands in our study area but the confidence intervals are relatively small with a range of only 0.08 between the lower and upper limits. We reviewed the sentence at l.196-200.

Reviewer 3 Report

I read carefully the paper entitled: ”Occupancy of the American Three-toed Woodpecker in a Heavily Managed Boreal Forest of Eastern Canada”.

Although the paper has a certain scientific level, we notice some weaknesses:

On the whole, the work has a relatively simplistic character, the results being intuitive almost from the beginning.

Part required - Literature Review (1.5 pages), after Introduction.

To see if the data in the Conclusions are written correctly (or if necessary) – L.283-286 (”providing evidence that heavy forest harvesting is a detrimental driver that likely contributed to the significant long-term declining trends of the species in the province of Québec [-0.96%.year-1; 95% C.I.: [- 0.70– -1.22]; ,50] and at a larger scale in the boreal hardwood transition bird conservation region [-3.5%.year-1; 95% C.I.: [-2.0 – -5.2]; ,51] over the 1970-2019 period. T”).  

The Conclusions must be original / proper to the authors. That is why I suggest that they be resynthesized and no longer contain extended references to the works of other authors. (50. Desrochers, A. Tendances ornithologiques du Québec Available online: https://www.toq.ffgg.ulaval.ca/ (accessed on Oct 13, 2020). 51. Meehan, T.D.; LeBaron, G.S.; Dale, K.; Michel, N.L.; Verutes, G.M.; Langham, G.M. Abundance trends of birds wintering in the USA and Canada, from Audubon Christmas Bird Counts, 1966-2017, version 2.1.; National Audubon Society: New York, NY, USA, 2018.

If they read carefully the text of the paper, the authors will detect for themselves the negligences ...

But he must apply the rules of the Journal.

It would be good to expand the bibliography with some articles from prestigious scientific journals (WoS) published in 2019 and 2020. 

Author Response

I read carefully the paper entitled: ”Occupancy of the American Three-toed Woodpecker in a Heavily Managed Boreal Forest of Eastern Canada”.

Although the paper has a certain scientific level, we notice some weaknesses:

On the whole, the work has a relatively simplistic character, the results being intuitive almost from the beginning.

Part required - Literature Review (1.5 pages), after Introduction.

***We don’t understand what exactly the reviewer recommended us to do here.

To see if the data in the Conclusions are written correctly (or if necessary) – L.283-286 (”providing evidence that heavy forest harvesting is a detrimental driver that likely contributed to the significant long-term declining trends of the species in the province of Québec [-0.96%.year-1; 95% C.I.: [- 0.70– -1.22]; ,50] and at a larger scale in the boreal hardwood transition bird conservation region [-3.5%.year-1; 95% C.I.: [-2.0 – -5.2]; ,51] over the 1970-2019 period. T”).  

***We confirm data in the conclusions are written correctly.

The Conclusions must be original / proper to the authors. That is why I suggest that they be resynthesized and no longer contain extended references to the works of other authors. (50. Desrochers, A. Tendances ornithologiques du Québec Available online: https://www.toq.ffgg.ulaval.ca/ (accessed on Oct 13, 2020). 51. Meehan, T.D.; LeBaron, G.S.; Dale, K.; Michel, N.L.; Verutes, G.M.; Langham, G.M. Abundance trends of birds wintering in the USA and Canada, from Audubon Christmas Bird Counts, 1966-2017, version 2.1.; National Audubon Society: New York, NY, USA, 2018.

***We used reference to support our conclusion, which seem appropriate to us, and our conclusions are original and proper to us.

If they read carefully the text of the paper, the authors will detect for themselves the negligences ...

***We don’t understand what exactly the reviewer recommended us to do here.

But he must apply the rules of the Journal.

***We applied the rules of the Journal as we understand them.

It would be good to expand the bibliography with some articles from prestigious scientific journals (WoS) published in 2019 and 2020. 

***We added more recent articles in the Discussion (see l.271-277).